# Temperature and Salt Responsive Amphoteric Nanogels Based on *N*-Isopropylacrylamide, 2-Acrylamido-2-methyl-1-propanesulfonic Acid Sodium Salt and (3-Acrylamidopropyl) Trimethylammonium Chloride

**DOI:** 10.3390/nano12142343

**Published:** 2022-07-08

**Authors:** Aigerim Ye. Ayazbayeva, Alexey V. Shakhvorostov, Iskander Sh. Gussenov, Tulegen M. Seilkhanov, Vladimir O. Aseyev, Sarkyt E. Kudaibergenov

**Affiliations:** 1Laboratory of Functional Polymers, Institute of Polymer Materials and Technology, Almaty 050019, Kazakhstan; alex.hv91@gmail.com (A.V.S.); iskander.gusenov@mail.ru (I.S.G.); 2Department of Chemical and Biochemical Engineering, Satbayev University, Almaty 050013, Kazakhstan; 3Laboratory of NMR-Spectroscopy, Sh. Ualikhanov University, Kokshetau 020000, Kazakhstan; tseilkhanov@mail.ru; 4Department of Chemistry, University of Helsinki, 00014 Helsinki, Finland; vladimir.aseyev@helsinki.fi

**Keywords:** polyampholyte nanogels, thermal response, salt response, phase transition temperature, hydrophobic/hydrophilic balance

## Abstract

Polyampholyte nanogels based on *N*-isopropylacrylamide (NIPAM), (3-acrylamidopropyl) trimethylammonium chloride (APTAC) and 2-acrylamido-2-methyl-1-propanesulfonic acid sodium salt (AMPS) were synthesized via conventional redox-initiated free radical copolymerization. The resultant nanogels of various compositions, specifically [NIPAM]:[APTAC]:[AMPS] = 90:5:5; 90:7.5:2.5; 90:2.5:7.5 mol.%, herein abbreviated as NIPAM_90_-APTAC_5_-AMPS_5_, NIPAM_90_-APTAC_7.5_-AMPS_2.5_ and NIPAM_90_-APTAC_2.5_-AMPS_7.5_, were characterized by a combination of ^1^H NMR and FTIR spectroscopy, TGA, UV–Vis, DLS and zeta potential measurements. The temperature and salt-responsive properties of amphoteric nanogels were studied in aqueous and saline solutions in a temperature range from 25 to 60 °C and at ionic strengths (μ) of 10^−3^ to 1M NaCl. Volume phase transition temperatures (VPTT) of the charge-balanced nanogel were found to reach a maximum upon the addition of salt, whereas the same parameter for the charge-imbalanced nanogels exhibited a sharp decrease at higher saline concentrations. A wide bimodal distribution of average hydrodynamic sizes of nanogel particles had a tendency to transform to a narrow monomodal peak at elevated temperatures and higher ionic strengths. According to the DLS results, increasing ionic strength results in the clumping of nanogel particles.

## 1. Introduction

Aqueous dispersions of particles swollen in water, which have a diameter of tens to hundreds of nanometers, formed by cross-linked polymer chains and having a three-dimensional porous structure, are defined as nanogels and widely used in medicine, wastewater treatment, catalysis, and the oil industry, among others [1,2]. The chemical structure of polymers used in the synthesis of nanogels is a hydrophilic framework with grafted hydrophobic fragments, which also provides such nanogels with the ability to swell or deswell [3]. The nanogels can respond to alterations in pH, temperature, salinity, solvent exchange, and other factors, which makes possible the control of their porosity, size, and strength [4]. A range of methods of synthesis can be used with various types of monomers in the creation of a number of nanogels with universal and multifunctional properties, as well as broad applications. Thermoresponsive nanogels with volume phase transition temperatures (VPTT), at which phase separation occurs, are of great interest [5,6]. A common polymer possessing thermal responsiveness in water is poly-N-isopropylacrylamide (PNIPAM), which has a lower critical solution temperature (LCST) of ~33 °C. Below this LCST, PNIPAM is hydrophilic and soluble in water, due to the hydrogen bonds formed between the amide groups and water, whereas, above the LCST, it is hydrophobic and insoluble in water, because of hydrophobic interactions between the isopropyl groups [7,8,9]. Hydrophobic isopropyl groups attached to hydrophilic amide groups, as well as the hydrophobic backbone, make the PNIPAM thermosensitive. Copolymerization of the monomer NIPAM with various other ionic monomers decreases or increases the LCST due to interference with hydrogen bonds between the amide groups of PNIPAM and water [10]. Stimuli-responsive zwitterionic polymers are of special interest [11]. Zwitterionic polymers contain both positive and negative charges within the same molecule, making them electrically neutral [12]. Zwitterionic nanogels have been described by Chen et al., who were able to synthesize monodisperse microgel particles based on NIPAM, acrylic acid and *N*-[3-(dimethylamino)propyl]methacrylamide, which undergo a reversible volume change in response to alterations in pH and temperature [13]. Particles in aqueous dispersions of zwitterionic microgels have a minimum hydrodynamic diameter at the isoelectric point (IEP). In addition, the research described the controlled uptake and release of ionic and non-ionic surfactants from these microgel particles.

Ogawa et. al. [14] described the synthesis of cross-linked nanogels based on anionic acrylic acid (AA) and cationic 1-vinylimidazole (VI) monomers into a chain of thermosensitive NIPAM, in AA/VI ratios of 1:1 and 1:4, via redox polymerization. The diameter of particles within the polyampholyte nanogels varied with pH, showing the smallest value at the IEP. AA/VI = 1:1, a nanogel with a balanced charge, demonstrated an antipolyelectrolyte effect at pH = IEP, due to inter-particle interaction.

The change in the radius of particles in NIPAM-based microgels containing 2% cationic groups, which were synthesized via precipitation polymerization in the presence of a cationic surfactant, cetyltrimethylammonium bromide, may depend on temperature and saline concentration. Upon addition of the ionic groups, phase transition occurs, raising the temperature from 33 to 44 °C. Conversely, with an increase in the ionic strength of the solution, the phase transition shifts towards lower temperatures, at which, due to charge competition, a salting-out effect occurs [15].

Microgels based on NIPAM and acrylic acid homologues (acrylic acid, methacrylic acid, 2-ethylacrylic acid, and 2-butylacrylic acid), using *N*,*N*′-methylenebisacrylamide (MBAA) as a crosslinking agent, are sensitive to changes in pH and temperature [16]. Microgels modified with 2-butyl acrylic acid have been found to exhibit unusually fast response kinetics to temperature changes, whereas those modified with acrylic acid show the slowest phase transition kinetics to temperature changes. This effect is explained by the different degrees of hydrophobicity of the acid groups; thus, increased hydrophobicity leads to a faster response to changes in the temperature of the medium.

The synthesis of a neutrally charged polyampholyte based on cationic vinylbenzyltrimethylammonium chloride (VBTAC) and anionic sodium p-styrenesulfonate (NaSS) via RAFT polymerization has been described by Kanta Sharker et al. [17]. The resulting copolymer is insoluble in water at room temperature but dissolves upon the addition of NaCl and a rise in temperature, thus showing phase behavior at the upper critical solution temperature (UCST). An increase in the concentration and molecular weight of the copolymer, as well as a decrease in the concentration of NaCl, raises the UCST.

Analysis of the applicable literature reveals that, despite the progress made in the field of synthesis and study of highly charged polyampholytes, there is a limited amount of information on highly charged thermo- and salt-sensitive polyampholyte nanogels. The current paper reports on the synthesis and characterization of novel charge-balanced and charge-imbalanced amphoteric nanogels based on NIPAM, (3-acrylamidopropyl) trimethylammonium chloride (APTAC) and 2-acrylamido-2-methyl-1-propanesulfonic acid sodium salt (AMPS). The behavior of these ternary polyampholytes in aqueous and saline solutions was studied to evaluate the conformational and phase transitions upon changes in temperature and salt content.

## 2. Materials and Methods

### 2.1. Materials

Polyampholyte nanogels were synthesized using the following chemicals: the monomers, N-isopropylacrylamide (NIPAM, 97% purity, Sigma-Aldrich Chemical Co., St. Louis, MO, USA), 2-acrylamido-2-methylpropanesulfonic acid sodium salt (AMPS, 50 wt.%, Sigma-Aldrich Chemical Co., St. Louis, MO, USA) and (3-acrylamidopropyl) trimethylammonium chloride (APTAC, 75 wt.%, Sigma-Aldrich Chemical Co., St. Louis, MO, USA); the polymerization initiator, ammonium persulfate (APS, 98% purity, Sigma-Aldrich Chemical Co., St. Louis, MO, USA); the crosslinking agent, N,N-methylenebis(acrylamide) (MBAA, 99% purity, Sigma-Aldrich Chemical Co., St. Louis, MO, USA), the surfactant, sodium dodecyl sulfate (SDS, Sigma-Aldrich Chemical Co., St. Louis, MO, USA). Further materials included sodium metabisulfite (SMBS, 97% purity, Sigma-Aldrich Chemical Co., St. Louis, MO, USA), sodium chloride (NaCl, Sigma-Aldrich Chemical Co., St. Louis, MO, USA) and dialysis tubing cellulose membrane (12–14 kDa, Sigma-Aldrich Chemical Co., St. Louis, MO, USA). All chemicals were used without further purification.

### 2.2. Synthesis of Nanogels Based on NIPAM-APTAC-AMPS

Polyampholyte nanogels were synthesized via conventional redox-initiated free radical copolymerization in the presence of a crosslinking agent at 80 °C for 4 h. NIPAM, APTAC, AMPS, MBAA and SDS were dissolved in deionized water at room temperature in a 100 mL beaker under constant stirring. The resulting solution was then filtered through a 5 µm syringe filter and purged with argon for 15–20 min in order to remove dissolved oxygen. The solution was then carefully transferred to a stoppered round bottom flask, and dry powders of APS and SMBS were added. The flask was immersed in a water bath of 80 °C under constant stirring. After 4 h, the flask was removed from the water bath and cooled to room temperature. The resulting nanogel solutions were subjected to dialysis in distilled water for 14 days. After dialysis nanogels were freeze-dried until moisture was completely removed. The dry nanogel powders obtained were used for further preparation of solutions. Ratios of NIPAM-APTAC-AMPS nanogels are given in Table 1.

### 2.3. Fourier-Transform Infrared Spectroscopy (FT-IR) of Nanogels Based on NIPAM-APTAC-AMPS

The functional groups of the NIPAM-APTAC-AMPS nanogels were characterized using Fourier-transform infrared spectroscopy, performed using a Cary 660 FTIR (Agilent, Santa Clara, CA, USA), equipped with pike MIRacle ATR accessory (attenuated total reflection mode). Before performing the measurement, the nanogel samples were freeze-dried for 24 h until moisture was completely removed. The spectra of the nanogels were measured in the wavenumber range of 500 to 4000 cm^−1^ at room temperature.

### 2.4. Thermogravimetric Analysis (TGA) of Nanogels Based on NIPAM-APTAC-AMPS

The thermal decomposition of the NIPAM-APTAC-AMPS nanogels was investigated using a LabSys Evo (Setaram, Caluire-et-Cuire, France) under a nitrogen atmosphere in a temperature range of 25 to 500 °C, with a heating rate of 10 °C min^−1^. The TG curve made possible the determination of the initial decomposition temperature and the residual mass percentage. Further, the maximum thermal decomposition temperature of the NIPAM-APTAC-AMPS nanogels was calculated using the differential peaks of the DTA curve.

### 2.5. Determination of Volume Phase Transition Temperatures of Nanogels Based on NIPAM-APTAC-AMPS

Aqueous and saline solutions of the NIPAM-APTAC-AMPS nanogels with concentrations from 0.1 to 0.5 wt.% are transparent at room temperature (25 °C). When the temperature is above the VPTT, the solution becomes milky white, and the transmittance decreases as the NIPAM-APTAC-AMPS nanogels dehydrate and become more hydrophobic, thus less soluble in water. The volume phase transition temperatures (VPTT), of the NIPAM-APTAC-AMPS nanogels in aqueous and saline solutions, with ionic strength (µ), equal to 0.001, 0.01, 0.1, 0.5 and 1 M NaCl, were determined by observing changes in the transmittance of the solution after changing the temperature. The volume phase transition temperatures of NIPAM-APTAC-AMPS nanogels in aqueous and saline solutions (at the ionic strength of the solution µ = 0.001; 0.01; 0.1; 0.5 and 1 M NaCl) correspond to the minimum points on the differential curves. The experiments were carried out on a UV–Vis spectrophotometer (Specord 210 plus, Jena, Germany) using a concentration of 0.1 wt.% NIPAM-APTAC-AMPS at λ = 700 nm and a 0.5 °C min^−1^ heating rate over a temperature range of 25–60 °C, as described in previous research by the current team [18].

### 2.6. Determination of the Average Hydrodynamic Size and Zeta Potential of Nanogels Based on NIPAM-APTAC-AMPS

The average hydrodynamic size (R_h_) and zeta potential (ζ) of the NIPAM-APTAC-AMPS nanogels in aqueous and saline solutions were measured using dynamic light scattering (DLS) on a Zetasizer Nano ZS90 (Malvern Instruments, Malvern, UK). The particle size was measured in a 0.1 wt.% solution of nanogels at a range of 25 to 50 °C at intervals of 5 °C and with µ = 0.001, 0.1 and 1 M NaCl. Zeta potential of the nanogels was measured first in an aqueous solution with 0.1 wt.% NIPAM-APTAC-AMPS at 25 °C, and then in a 0.001 M NaCl solution, which was heated from 25 to 60 °C at intervals of 5 °C.

### 2.7. ^1^H NMR Spectroscopy of Nanogels Based on NIPAM-APTAC-AMPS

^1^H NMR spectra were recorded on a JNN-ECA 400 spectrometer (JEOL, Tokyo, Japan) at a frequency of 399.78 MHz, using D_2_O as a solvent. The temperature varied from 20 to 55 °C at intervals of 5 °C. For each temperature, the sample was allowed to stabilize for 10 min prior to measurement. Chemical shifts were measured relative to residual proton signals from the deuterated solvent.

### 2.8. SEM Data of Nanogels Based on NIPAM-APTAC-AMPS

In order to determine the surface morphology, 0.1 wt.% solutions of nanogels were preliminarily dried at 25 °C, and then analyzed using a MIRA 3 LMU scanning electron microscope (Tescan, Brno, Czech Republic).

## 3. Results

### 3.1. Synthesis and Characterization of Nanogels Based on NIPAM-APTAC-AMPS

NIPAM_90_-APTAC_5_-AMPS_5_, NIPAM_90_-APTAC_7.5_-AMPS_2.5_ and NIPAM_90_-APTAC_2.5_-AMPS_7.5_ were synthesized via conventional redox-initiated free radical copolymerization (Figure 1).

Based on research by Braun et al. [19] the composition of the NIPAM-APTAC-AMPS water-soluble terpolymers was expected to be insignificantly deviated (±0.5 mol.%) compared to the monomer composition in the feed, and the samples obtained were shown to have randomly distributed charges along the macromolecular chain.

The broad absorption band in the region of 3290–3500 cm^–1^ corresponds to the secondary and tertiary amine groups, and the absorption bands in the region of 2800–3000 cm^–1^ correspond to the asymmetric and symmetric vibrations of CH groups. The absorption bands at 1640 and 1540 cm^−1^ belong to the vibrations of the N-substituted groups, which are the amide I and amide II groups, respectively. The absorption band at 1460 cm^−1^ is characteristic of the bending vibrations of the CH groups. Finally, the absorption band in the region of 1040 cm^−1^ corresponds to the fluctuations of the S=O groups of the AMPS units (Figure 2).

### 3.2. SEM Analysis of the Nanogels

SEM data show that nanogel particles have glued or stuck aggregates connected by narrow necks and voids of nanometer and micron size (Figure 3). Due to multiple contacts between spherical or wormlike nanogels, macroscopic gelation occurs, leading to the formation of three-dimensional networks of nanogels [20].

### 3.3. TGA and DTA Data for NIPAM-APTAC-AMPS Nanogels

The results of thermogravimetric and differential thermal analysis for nanogels are shown in Figure 4.

The TGA curves demonstrate that three losses in weight occur. The first, at >100 °C, is probably due to the evaporation of physically adsorbed moisture. The second, at >300 °C, is due to the decomposition of NIPAM fragments. Complete thermal decomposition of the nanogels occurs in the temperature range of 410–420 °C.

### 3.4. Zeta Potentials (ζ) of NIPAM-APTAC-AMPS Nanogels in Aqueous and Saline Solutions

The zeta potentials of charge-balanced and charge-imbalanced solutions of 0.1 wt.% NIPAM-APTAC-AMPS polyampholyte nanogels were measured at 25 °C. In an aqueous solution, NIPAM_90_-APTAC_7.5_-AMPS_2.5_, with an excess of positively charged APTAC monomer, has ζ = +4 ± 1 mV. NIPAM_90_-APTAC_2.5_-AMPS_7.5_, with an excess of negatively charged AMPS monomer, has ζ = −7 ± 1 mV. In the case of the charge-balanced NIPAM_90_-APTAC_5_-AMPS_5_, the zeta potential is slightly positive, ζ = +1.5 ± 1 mV. Likely, this is due to a small excess of positively charged APTAC monomers.

The temperature dependence of the zeta potentials of the nanogels in 0.001 M NaCl solution is shown in Figure 5. In a temperature range of 25 °C to 60 °C, the charge-balanced nanogels have a zeta potential value of +4 ± 2 mV. However, in the case of charge-imbalanced nanogels, the zeta potential rose after the temperature of the solution had been increased. Thus, the positively charged nanogel, NIPAM_90_-APTAC_7.5_-AMPS_2.5_, has a zeta potential of around +5 mV at a temperature of 25 °C, but as the temperature rises to 40 °C, the zeta potential increases up to +25 mV. On the other hand, in the same temperature range, the negatively charged nanogel, NIPAM_90_-APTAC_2.5_-AMPS_7.5_, displayed a monotonic change in the zeta potential from −5 mV up to −30 mV. At 25 °C, the nanogels are thought to be predominantly in the swollen coiled state. With an increase in temperature, hydrophobic interactions between isopropyl radicals are enhanced, causing compactization of coils and the progressive release of charged groups to the outer layers of these compressed coils. This leads to the accumulation of positive or negative charges on the outside of charge-imbalanced nanogel particles and an increase in the zeta potential of the macromolecules. For charge-balanced nanogels, this effect also takes place, but due to mutual compensation of positively and negatively charged APTAC and AMPS groups, the zeta value changes insignificantly.

### 3.5. The Volume Phase Transition Behavior of Charge-Balanced NIPAM_90_-APTAC_5_-AMPS_5_ Nanogel in Saline Solutions

Figure 6 and Figure 7 and Table 2 show the effect of temperature and salinity on the volume phase transition temperature (VPTT) of the charge-balanced nanogel, NIPAM_90_-APTAC_5_-AMPS_5_. At lower ionic strengths, μ = 0.001 and 0.01 M NaCl, the VPTT is slightly shifted to a higher temperature. The transmittance curves at μ = 0.001 Mand 0.01 M are equal to 39.2 °C and 40.4 °C, respectively. The reason for this phenomenon is that at μ = 0.001 and 0.01 the ionic contacts formed between the APTAC and AMPS monomers are gradually disrupted due to screening of the electrostatic attraction between oppositely charged moieties by low-molecular-weight salts, demonstrating the antipolyelectrolyte effect. The sense of the antipolyelectrolyte effect is unfolding (or swelling) of macromolecular chains of annealed (pH-dependent) polyampholytes at the isoelectric point (IEP) and/or charge-balanced (pH-independent) polyampholytes at quasi-neutral state in salt solution due to the screening of the electrostatic attraction between oppositely charged moieties. The antipolyelectrolyte effect was confirmed by experimental results of authors [21,22,23] obtained for both annealed and quenched polyampholytes in solution and gel state. Earlier, the same research team showed that the intrinsic viscosity of charged-balanced polyampholyte APTAC_50_-co-AMPS_50_ increases in saline solution, another example of the antipolyelectrolyte effect [24]. The behavior of charge-balanced amphoteric nanogel NIPAM_90_-APTAC_5_-AMPS_5_ at the IEP (or in a quasi-neutral state) is in good agreement with the theory of Khokhlov et al. [25], e.g., the swelling of polyampholyte gel near the IEP should increase with the increase in the ionic strength of the solution. For this reason, in the case of NIPAM_90_-APTAC_5_-AMPS_5_, the liberated ionic pairs of APTAC and AMPS contribute to better solubility by increasing the hydrophilicity of the system. As the ionic strength of the solution increases further, the VPTT shifts up to 47.8 °C and is maximal at μ = 0.1 M. At an extremely high ionic strength, μ = 0.5 and 1 M, the values of the VPTT decrease and are equal to 45.1 and 38.3 °C. In our mind, at µ = 0.5 and 1 M, the number of the counterions is sufficient to completely screen charges of both signs, thus the charged macromolecules approach neutrality. Moreover, at a high salt concentration, µ = 0.5–1.0 M, the salting-out effect likely occurs with respect to ionic groups, leading to decreasing in the volume phase transition temperature. Under such conditions, the solubility and phase behavior of NIPAM_90_-APTAC_5_-AMPS_5_ is probably determined mostly by NIPAM_90_ fragments. The influence of the NaCl concentration on the behavior of linear and crosslinked PNIPAM as a function of temperature was studied by authors [26,27]. It was established that both the LCST of linear PNIPAM and VPPT of crosslinked PNIPAM decrease as the concentration of NaCl increases. Thus, the volume phase transition behavior of charge-balanced NIPAM_90_-APTAC_5_-AMPS_5_ nanogel with respect to salt and temperature has a complex character. The addition of NaCl tends to increase the hydrophilicity of ionic groups due to the antipolyelectrolyte effect and increase the hydrophobicity of NIPAM due to the enhancement of inter- and intramolecular hydrophobic interactions. The increasing temperature enhances the entropic contribution to the free energy augments and decreases the quality of water with respect to NIPAM_90_ moieties causing the phase separation.

Of note is that less turbidity change occurs with NIPAM_90_-APTAC_5_-AMPS_5_ at μ > 0.1 M NaCl, than happens when the nanogel is in solution at lower salinities (Figure 6). The size and density of individual particles affect scattering, which is then even more altered by elevated temperatures, resulting in clumping, which in turn changes transmittance. This suggests that the particles formed in solutions with μ > 0.1 M are smaller than those formed at lower ionic strengths, which has been confirmed by the dynamic light scattering data (Figure 8).

### 3.6. The Mean Hydrodynamic Size of Charge-Balanced NIPAM_90_-APTAC_5_-AMPS_5_ Nanogel in Saline Solutions

Distributions of the mean hydrodynamic radius (R*_h_*) were studied for the NIPAM_90_-APTAC_5_-AMPS_5_ nanogel within a temperature range of 25–50 °C and at μ = 0.001, 0.1 and 1 M NaCl (Figure 8). At 25 °C and μ = 0.001 M, NIPAM_90_-APTAC_5_-AMPS_5_ has a particle size of R*_h_* = 18–20 nm (Figure 8a). Several peaks corresponding to ~10, 40 and 90 nm are observed in solutions at higher ionic strengths (μ = 0.1 and 1.0). The appearance of such large particles is probably connected with the unfolding of macromolecular chains, due to the antipolyelectrolyte effect, and the formation of clumps of nanogel particles. However, such clumps are not numerous.

The unfolding of nanogel particles is promoted when the temperature is increased. In the transition region, T = 35–40 °C, the R*_h_* of the clumps reached 500–600 nm at μ = 0.001 M NaCl. Under these conditions, the antipolyelectrolyte effect is negligible, and the hydrophobicity of the NIPAM units dominates. A volume phase transition occurs at 47.8 °C and μ = 0.1 M. Unfolding of the nanogel particles increases as the temperature is raised, up to a maximum at 45.1 °C, but then the structure of the particles remains constant (Figure 8b). Further increases in ionic strength result in the electrostatic screening of the APTAC and AMPS units. The antipolyelectrolyte effect is suppressed at μ = 1 M, at which point volume phase transition is solely determined by the NIPAM units (Figure 8c). In all the solutions studied, clumping in the nanogels remained colloidally stable above the VPTT, and no macroscopic phase separation was observed.

### 3.7. The Volume Phase Transition Behavior of NIPAM_90_-APTAC_7.5_-AMPS_2.5_ and NIPAM_90_-APTAC_2.5_-AMPS_7.5_ Nanogels in Saline Solutions

Figure 9 presents the dependence of transmittance on one axis and temperature on the other for charge-imbalanced nanogels. The NIPAM_90_-APTAC_7.5_-AMPS_2.5_ nanogel contains an excess of positively charged APTAC monomers. According to the theory of Khokhlov et al. [25] for the gel with small excess of the charges of one sign, more complex behavior should be observed. At low concentrations of salt, the network should shrink, and on further increase in the ionic strength, it should swell. This statement may be interpreted as follows. Quantitatively, NIPAM_90_-APTAC_7.5_-AMPS_2.5_ nanogel can be considered as a combination of NIPAM_90_, a neutral part, APTAC_2.5_-AMPS_2.5_, the charge-balanced polyampholyte part, and APTAC_5_, the positively charged polyelectrolyte part. The addition of salt tends to cause the unfolding of the charge-balanced part, APTAC_2.5_-AMPS_2.5_, due to the antipolyelectrolyte effect. At the same time, the added salt tends to diminish the positively charged part, APTAC_5_, because of the screening of the electrostatic repulsion between the positively charged APTAC_5_ groups, demonstrating the polyelectrolyte effect. In other words, the conformation of the charge-balanced part, APTAC_2.5_-AMPS_2.5_, can be considered the core, and the positively charged part, APTAC_5_, the shell, where the core acts like a polyampholyte, while the shell functions like a polyelectrolyte. The addition of low-molecular-weight salts tends to shrink the shell and swell the core, mutual compensation of which, in the range of μ = 0.001, 0.01 and 0.1 M NaCl, likely leads to the VPTT for NIPAM_90_-APTAC_7.5_-AMPS_2.5_ remaining more or less constant, at 45, 44.4 and 44.3 °C, respectively (Table 2). At higher ionic strengths, μ = 0.5 and 1 M, the antipolyelectrolyte effect (swelling) prevails over the polyelectrolyte effect (shrinking) and leads to the swelling of the nanogel. This is consistent with the theory of Khokhlov et al. [25]. As stated in Section 3.5, at µ = 0.5 and 1 M the solubility and phase behavior of NIPAM_90_-APTAC_7.5_-AMPS_2.5_ is probably determined mostly by NIPAM_90_ fragments. Due to temperature-dependent enhancement of inter- and intramolecular hydrophobic interactions between NIPAM_90_ fragments the VPTT is shifted to lower temperatures, 39.5 and 33.5 °C, respectively (Table 2). NIPAM_90_-APTAC_2.5_-AMPS_7.5_ contains an excess of negatively charged AMPS, so its behavior is similar to NIPAM_90_-APTAC_7.5_-AMPS_2.5_. However, the interpretation of the volume phase behavior of NIPAM_90_-APTAC_2.5_-AMPS_7.5_, with respect to the combined action of temperature and salinity, is complicated and requires further detailed research. The transmittance curves for NIPAM_90_-APTAC_2.5_-AMPS_7.5_ are quite different from NIPAM_90_-APTAC_7.5_-AMPS_2.5_, and at high ionic strengths, μ = 0.5 and 1 M, contain two volume phase transition temperatures, VPTT = 51.1 and 47.1 °C, respectively (Table 2).

### 3.8. The Mean Hydrodynamic Size of Charge-Imbalanced NIPAM_90_-APTAC_7.5_-AMPS_2.5_ and NIPAM_90_-APTAC_2.5_-AMPS_7.5_ Nanogels in Saline Solutions

Figure 10 presents the effect of temperature, ranging from 25 to 50 °C, and ionic strength (μ) on the hydrodynamic nanogel particle size of NIPAM_90_-APTAC_7.5_-AMPS_2.5_. The mean average, R*_h_*, is ~10 nm for the NIPAM_90_-APTAC_7.5_-AMPS_2.5_ nanogel in solutions where μ < 0.1 M NaCl at room temperature. R*_h_* increases to 100–110 nm when the temperature is raised to 40 °C. However, in solutions with an ionic strength of 1 M, even at room temperature, 25 °C, a larger average hydrodynamic size was observed, compared to those with a lower μ, though an increase in temperature did not cause a major change in size.

NIPAM_90_-APTAC_2.5_-AMPS_7.5_ did not demonstrate strict regularity in terms of average hydrodynamic radius at room temperature, 25 °C, and clumping of nanogel particles could already be observed. At higher temperatures, sizes gradually increased. The R*_h_* dramatically rose above VPTT, due to the formation of multi-particle clumps. Small ones, of 10–20 nm, were detected below the VPTT [28]. Figure 11 represents the effect of temperature, from 25 to 50 °C, and ionic strength on the hydrodynamic size of NIPAM_90_-APTAC_7.5_-AMPS_2.5_.

**Figure 10 nanomaterials-12-02343-f010:**
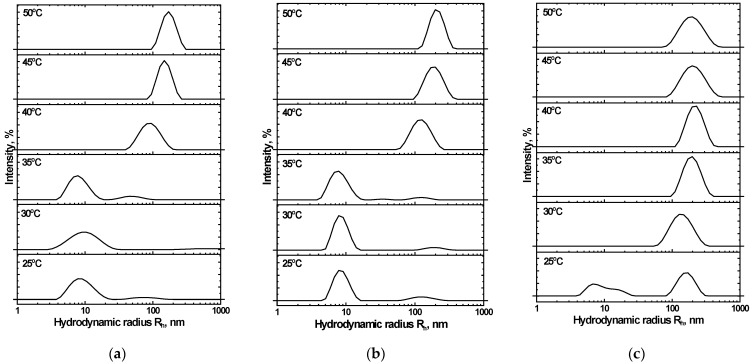
Effect of temperature and ionic strength (µ) on the mean hydrodynamic radius (R*_h_*) of NIPAM_90_-APTAC_7.5_-AMPS_2.5_: µ = (**a**) 0.001, (**b**) 0.1 and (**c**) 1.0.

### 3.9. Temperature-Variable ^1^H NMR Analysis

The composition of the nanogels obtained was determined by means of proton nuclear magnetic resonance spectroscopy, using D_2_O as a solvent. Chemical shifts, δ, were given in ppm. To quantitatively describe the degree of the volume phase transition of NIPAM-APTAC-AMPS, ^1^H NMR measurements were performed as a function of temperature (Figure 12 and Figure 13). Almost all the protons of the PNIPAM units can be observed in the ^1^H NMR spectra and assigned. However, the exact number of APTAC and AMPS units cannot be precisely determined, due to the overlapping of certain peaks (Figure 12).

The resonance peaks *a* and *b* are attributed to PNIPAM segments, with peak *b* chosen as the reference to calculate the degree of volume phase transition of the NIPAM-APTAC-AMPS nanogels. The D_2_O peak does not shift over the whole temperature range of 25 to 55 °C. Though, as expected, all other peaks shifted toward the lower field, and a notable decrease in intensity was observed as the temperature was raised [29,30]. The proton signals of PNIPAM moieties after volume phase transition were barely detectable, and only the protons from the surface of the nanogel particles, still soluble in D_2_O, could be detected.

**Figure 13 nanomaterials-12-02343-f013:**
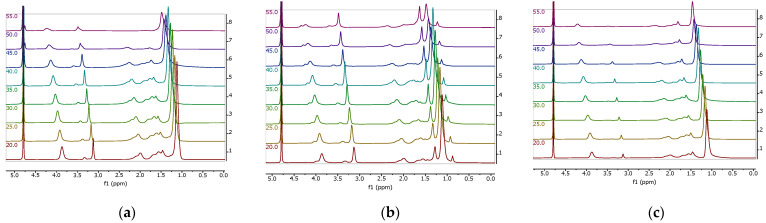
Temperature-variable ^1^H NMR of NIPAM-APTAC-AMPS nanogels: (**a**) NIPAM_90_- APTAC_5_-AMPS_5_; (**b**) NIPAM_90_-APTAC_7.5_-AMPS_2.5_; (**c**) NIPAM_90_-APTAC_2.5_-AMPS_7.5_.

By using Equation (1), characterization of the degree of volume phase transition is possible:(1)$$p=1-\left(\frac{I}{I_{0}}\right),$$
where *p* is volume phase-separated fraction, and *I* and *I*_0_ are the integrated intensities of peak *b* at the specified temperature and at 20 °C, respectively.

Figure 14 shows how changing temperature influences the volume phase-separated fraction, *p,* for different compositions of NIPAM-APTAC-AMPS nanogels, which exhibit a much weaker volume phase transition than free linear PNIPAM, due to the strongly charged APTAC and AMPS groups along the macromolecular chain. At higher temperatures, the hydrophobization of NIPAM groups can create a hydrophobic core, while the ionic APTAC and AMPS groups can stabilize the nanogel particles in an aqueous medium. Therefore, permeability, stability and dispersion of drug-loaded amphoteric nanogels can potentially be used in drug delivery systems [30]. 

Following volume phase transition, NIPAM-APTAC-AMPS nanogels demonstrate a higher volume phase-separated fraction, from 0.54 to 0.72, which is much lower than that of linear pure PNIPAM, *p* = 0.95. The introduction of APTAC and AMPS groups can be concluded to be responsible for decreases in volume phase transition temperature. For precise determination of volume phase transition temperatures, Boltzmann fitting was calculated using Equation (2) for all points in Figure 14.
(2)$$y=\frac{A_{1}-A_{2}}{1+e^{\frac{\left(x-x_{0}\right)}{dx}}}+A_{2},$$
where *A*_1_ is the minimum value of the function; *A*_2_ is the maximum value of the function; *x*_0_ is the value on the *x* axis corresponding to the inflection of the curve, which corresponds to the volume phase transition temperature, VPTT, and *dx* is the domain where x_0_ lies [31].

Charge-balanced NIPAM_90_-APTAC_5_-AMPS_5_ has a transition temperature of around 48.7 °C, but for the charge-imbalanced, positively and negatively, nanogels the transition temperatures are 44.8 and 48.7 °C, respectively. The volume phase transition temperature of the nanogels, as determined by UV–Vis in deionized water, is equal to 40.4 °C for the charge-balanced gel, 41.5 °C for the negatively charged variant, and 44.1 °C for the positively charged one. These values are lower than calculated using Equation (1), because in the volume phase separation fraction, the presence of ionic APTAC and AMPS are thought to weaken the volume phase transition behavior of the nanogels.

## 4. Conclusions

In the charge-balanced nanogel, NIPAM_90_-APTAC_5_-AMPS_5_, the oppositely charged monomers mutually compensate each other in pure water. However, the volume phase transition temperature (VPTT) of NIPAM_90_-APTAC_5_-AMPS_5_ increases upon the addition of salt, which can be explained by the antipolyelectrolyte effect. At the lower and moderate ionic strength, µ = 0.001–0.1 M NaCl, the electrostatic attraction between positively and negatively charged monomers gradually decreases, which leads to the unfolding of polymer chains, additional hydrophilization, and an increase in the VPTT. At the high ionic strengths, μ = 0.5 and 1 M NaCl, the macromolecular system becomes similar to a neutral polymer, because the salt ions completely screen the positive and negative charges of the macromolecules. Under such conditions, the solubility and volume phase behavior of NIPAM_90_-APTAC_5_-AMPS_5_ is likely determined by the NIPAM fragments. Moreover, the higher the ionic strength, the more often the salting-out effect occurs, leading to overall hydrophobization of the macromolecular chains and, consequently, to a decrease in the VPTT.

The positively charged nanogel, NIPAM_90_-APTAC_7.5_-AMPS_2.5_, was shown to be in an expanded state in pure water. However, the nanogel particles shrink as the salt concentration increases, µ = 0.5–1.0 M NaCl, due to the screened electrostatic repulsion between uniformly charged macroions, demonstrating the polyelectrolyte effect, leading to sharp decreases in the VPTT, which was shown to be less sensitive to temperature in the range of µ = 0.001–0.1 M NaCl, but sharply decreased at the higher salt concentrations. An attempt has been made to interpret this phenomenon in terms of core–shell competition, in which a positively charged core acts like a polyampholyte and a negatively charged shell functions like a polyelectrolyte, as salinity is increased. The negatively charged nanogel, NIPAM_90_-APTAC_2.5_-AMPS_7.5_, with respect to the combined action of temperature and salt, is complicated and requires additional research. 

Further, the volume phase transition of the NIPAM-APTAC-AMPS nanogels in pure deuterated water was studied using ^1^H NMR as a function of temperature. A conclusion was made that the introduction of small amounts of APTAC and AMPS groups into NIPAM chains is responsible for the increase in the VPTT of the resultant nanogels in comparison with that of pure PNIPAM. At higher temperatures, the hydrophobization of NIPAM groups can create a hydrophobic core, while the ionic APTAC and AMPS groups stabilize the nanogel particles in an aqueous medium. According to Equation (1), which utilized the NMR results, the VPTT of NIPAM_90_-APTAC_5_-AMPS_5_ was around 48.7 °C, whereas for NIPAM_90_-APTAC_7.5_-AMPS_2.5_ and NIPAM_90_-APTAC_2.5_-AMPS_7.5_ the values were to 44.8 and 48.7 °C, respectively. However, when using UV–Vis in deionized water, the VPTT of the nanogels was found to be equal to 40.4 °C for NIPAM_90_-APTAC_5_-AMPS_5_, 44.1 °C for NIPAM_90_-APTAC_7.5_-AMPS_2.5_ and 41.5 °C for NIPAM_90_-APTAC_2.5_-AMPS_7.5_. These results are lower than those calculated using Equation (1), which is probably connected to the accuracy of the two methods.

## Figures and Tables

**Figure 1 nanomaterials-12-02343-f001:**
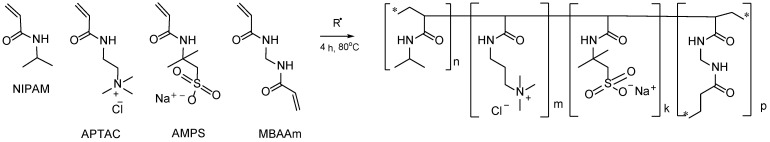
Schematic representation of free radical copolymerization of NIPAM, APTAC and AMPS monomers for nanogel synthesis.

**Figure 2 nanomaterials-12-02343-f002:**
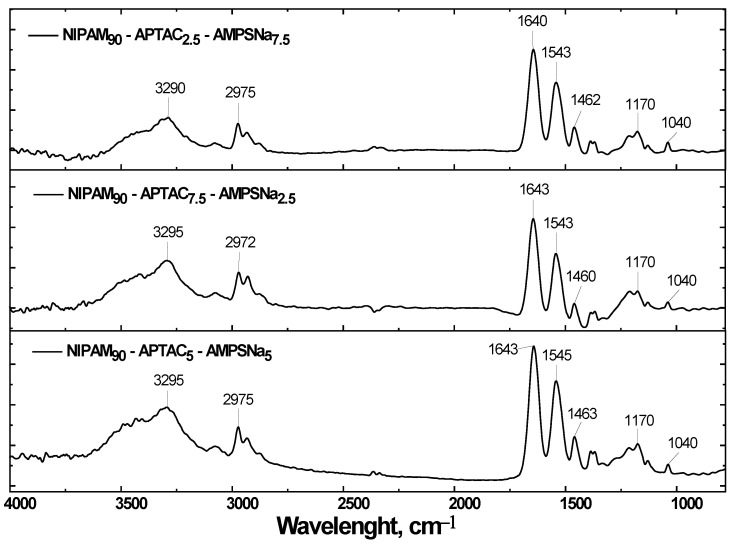
FTIR spectra of nanogels based on NIPAM-APTAC-AMPS.

**Figure 3 nanomaterials-12-02343-f003:**
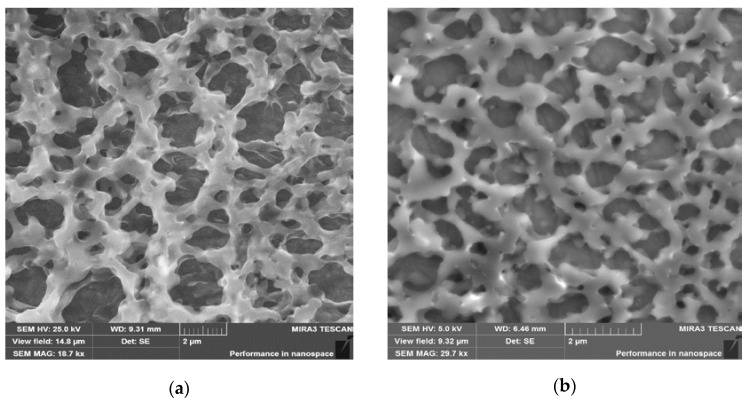
SEM images of (**a**) NIPAM_90_-APTAC_7.5_-AMPS_2.5_ and (**b**) NIPAM_90_-APTAC_2.5_-AMPS_7.5_ nanogels.

**Figure 4 nanomaterials-12-02343-f004:**
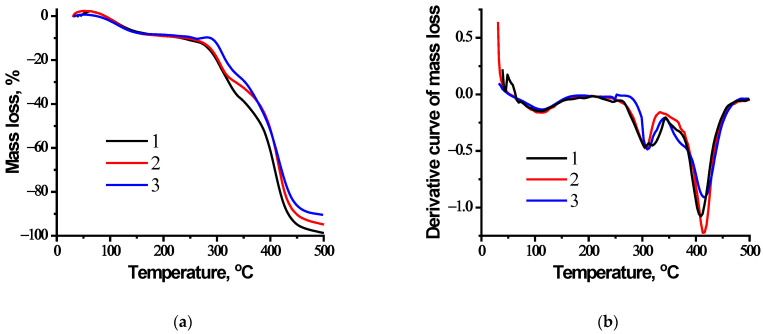
(**a**) TGA thermograms and (**b**) differential curves of nanogels, with curves (1) NIPAM_90_-APTAC_5_-AMPS_5_; (2) NIPAM_90_-APTAC_7.5_-AMPS_2.5_; (3) NIPAM_90_-APTAC_2.5_-AMPS_7.5_.

**Figure 5 nanomaterials-12-02343-f005:**
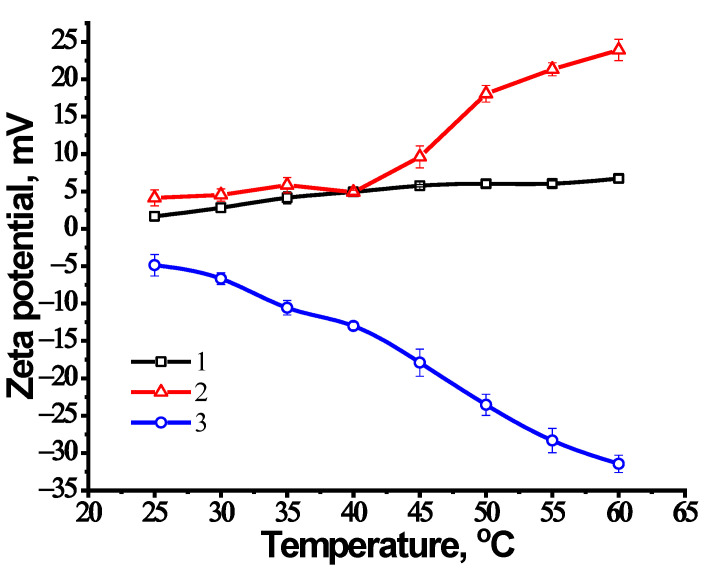
Zeta potential of nanogels in 0.001 M NaCl solution, with curves (1) NIPAM_90_-APTAC_5_-AMPS_5_; (2) NIPAM_90_-APTAC_7.5_-AMPS_2.5_; (3) NIPAM_90_-APTAC_2.5_-AMPS_7.5_.

**Figure 6 nanomaterials-12-02343-f006:**
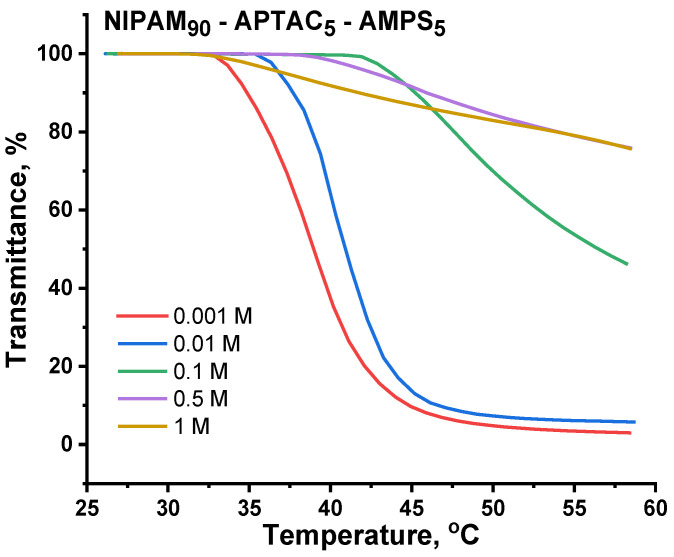
Effect of temperature and ionic strength (μ) on transmittance of the nanogel NIPAM_90_-APTAC_5_-AMPS_5_.

**Figure 7 nanomaterials-12-02343-f007:**
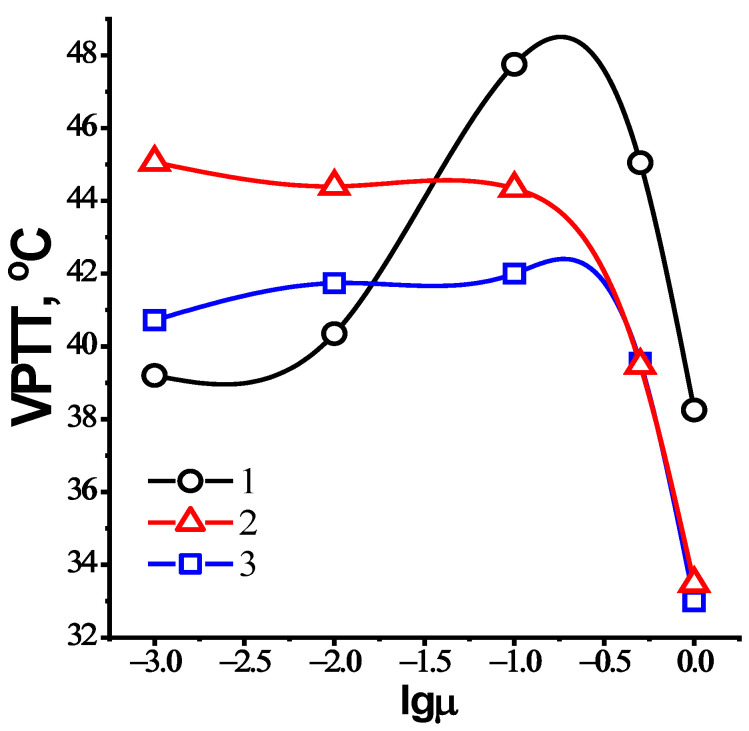
Temperature dependent phase behavior of nanogels based on NIPAM-APTAC-AMPS at various ionic strength (μ), with curves (1) NIPAM_90_-APTAC_5_-AMPS_5_; (2) NIPAM_90_-APTAC_7.5_-AMPS_2.5_; (3) NIPAM_90_-APTAC_2.5_-AMPS_7.5_.

**Figure 8 nanomaterials-12-02343-f008:**
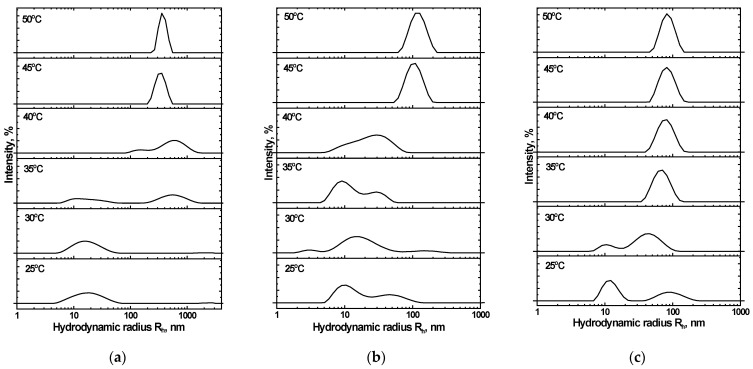
Effect of temperature and ionic strength (µ) on the mean hydrodynamic radius (R*_h_*) of NIPAM90-APTAC5-AMPS5: µ = (**a**) 0.001, (**b**) 0.1 and (**c**) 1.0.

**Figure 9 nanomaterials-12-02343-f009:**
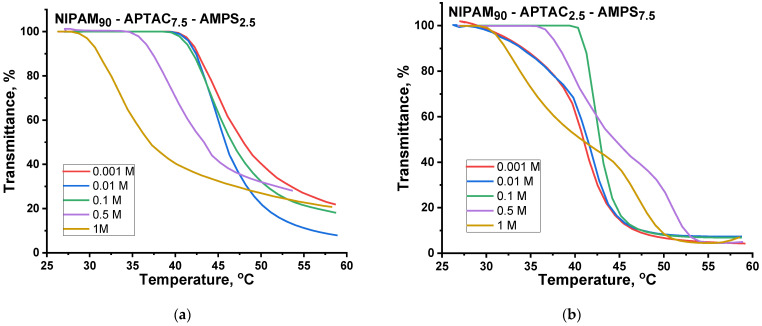
Effect of temperature and ionic strength (μ) on transmittance of (**a**) NIPAM_90_-APTAC_7.5_-AMPS_2.5_ and (**b**) NIPAM_90_-APTAC_2.5_-AMPS_7.5_ solutions.

**Figure 11 nanomaterials-12-02343-f011:**
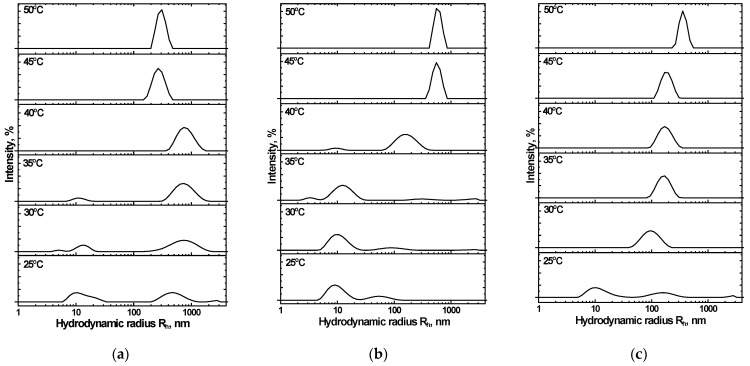
Effect of temperature and ionic strength (µ) on the mean hydrodynamic radius (R_h_) of NIPAM_90_-APTAC_2.5_-AMPS_7.5_: µ = (**a**) 0.001, (**b**) 0.1 and (**c**) 1.0.

**Figure 12 nanomaterials-12-02343-f012:**
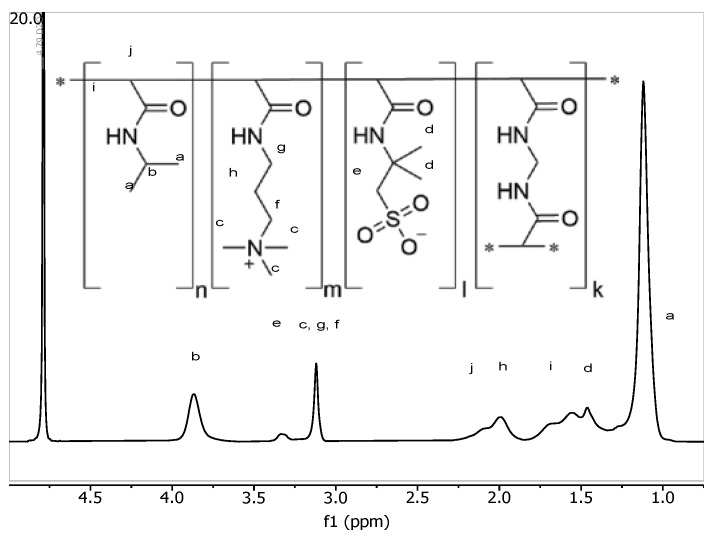
Temperature-variable ^1^H NMR of NIPAM-APTAC-AMPS nanogels.

**Figure 14 nanomaterials-12-02343-f014:**
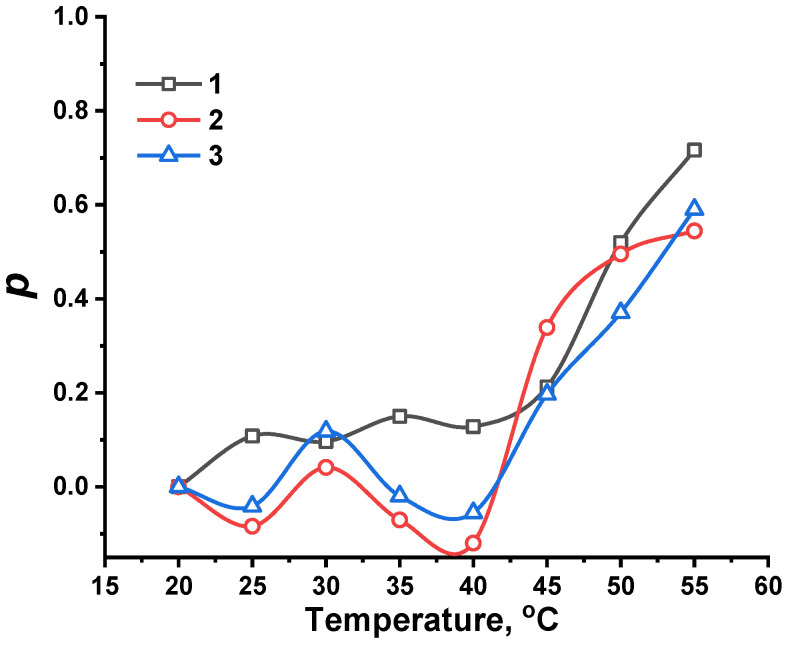
Temperature-dependent volume phase-separated fraction (*p*) for NIPAM-APTAC-AMPS nanogels in D_2_O, with curves (1) NIPAM_90_-APTAC_5_-AMPS_5_; (2) NIPAM_90_-APTAC_7.5_-AMPS_2.5_; (3) NIPAM_90_-APTAC_2.5_-AMPS_7.5_.

**Table 1 nanomaterials-12-02343-t001:** Ratios of NIPAM-APTAC-AMPS nanogels.

Initial Monomer Feed, mol. %			NIPAM, g	APTAC, g	AMPS, g	H_2_O, mL	APS, mg	SMBS, mg	MBAA, g	SDS, g	Yield, wt. %
NIPAM	APTAC	AMPS									
90	5	5	0.735	0.099	0.165	98.5	30	10	0.11	0.23	70
90	7.5	2.5	0.735	0.149	0.082	98.5	20	10	0.11	0.35	88
90	2.5	7.5	0.735	0.049	0.248	98.5	30	10	0.11	0.23	67

**Table 2 nanomaterials-12-02343-t002:** The effect of ionic strength (μ) on the volume phase transition temperature of NIPAM-APTAC-AMPS nanogels.

Nanogel	Ionic Strength μ, mol⋅L^−1^ (NaCl)					
	0	0.001	0.01	0.1	0.5	1.0
	Volume Phase Transition Temperature VPTT, °C					
NIPAM_90_-APTAC_5_-AMPS_5_	40.4	39.2	40.4	47.8	45.1	38.3
NIPAM_90_-APTAC_7.5_-AMPS_2.5_	44	45.0	44.4	44.3	39.5	33.5
NIPAM_90_-APTAC_2.5_-AMPS_7.5_	41.5 -	40.7 -	41.7 -	42.0 -	39.6 51.1 ^1^	33.0 47.1 ^1^

^1^ the second VPTT.

## Data Availability

The data presented in this study are available in this article.

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
