# Peer review of "Temperature and Salt Responsive Amphoteric Nanogels Based on N-Isopropylacrylamide, 2-Acrylamido-2-methyl-1-propanesulfonic Acid Sodium Salt and (3-Acrylamidopropyl) Trimethylammonium Chloride"

_nanomaterials, 2022, doi:10.3390/nano12142343_

Round 1

Author Response

The authors of the manuscript thank the reviewer for very positive and helpful comments, which have been taken into account in the revised text. We were pleased to hear that our results provided interest. Please see our responses to the reviewer comments in the attachment.

Reviewer 2 Report

The article is devoted to the study of nanoparticles of amphoteric gels at different ratios of charged groups. The article is interesting and can be published. However, first of all the authors need to answer a number of questions.

- In Figure 4b, the derivative curves of mass loss either have a rather large positive maximum at low temperatures or non-monotonic behavior. I wonder if this has some physical meaning or it is just the rough processing of data shown in Figure 4a.

- Figure 5, I believe that Zet-potental of unbalanced nanogels is non-monotonic function of temperature and salt content and that finally, at sufficiently high temperature, it goes to zero. Could you please check it?

- Figure 6, it is seen that curve M=1 lies below curve M=0.5.  How can you explain this?

- Page 7, What do you mean by the anti-polyelectrolyte effect? And in general, is it possible to expect polyelectrolyte effects in gels with balanced bound charges of different signs. Khokhlov, et all (Adv. Polym. Sci. 1993, v.109, pp. 123-171) showed that the reaction of polyampholytic gels to the addition of salt differs from the reaction of polyelectrolytes, i.e. macromolecules carrying charges of the same name. For example, the former can absorb a significant amount of salt. The non-monotonicity in the behavior of gels with an uncompensated bound charge is due to the balance of effects characteristic for different limiting cases. With theory described in Khokhlov, et all Adv. Polym. Sci. 1993, v.109, pp. 123-171 and references herein, one can possible explain the nonmonotonic behavior without involving core-shell hypothesis.

- Figure 7, there is no reference to it in the text and its meaning is unclear.

- Figure 8 shows a number of double-humped curves, which may mean the coexistence of gels of different sizes (say, Fig.8b, 35C), and perhaps on different (coil and globule)  states, and that the transition between them is like a phase transition of the first order. Could you please comment on this.

- Fig. 13, the transition  was characterized by degree of phase transition, calculated from the relative height of the b peaks. But for gels with an uncompensated charge, the intensities of b peaks are very small and apparently do not change in the temperature range under consideration.  In addition, authors do not compare the obtained data on the transition temperature with those that can be extracted from the other experimental data, such as distribution over hydrodnamic radius

-  Typo in the sentence : Charge-balanced NIPAM90-APTAC5-AMPS5 has a transition temperature of 48. 7 C, but for the charge-imbalanced, positively and negatively, nanogels the transition temperatures are 44.8 and 48.7

Author Response

(The authors gave the same response as above.)

Round 2

Reviewer 1 Report

The authors took into account the comments, thus  the article may be published.